# A Bibliometric Analysis and Global Trends in Fascioliasis Research: A Neglected Tropical Disease

**DOI:** 10.3390/ani11123385

**Published:** 2021-11-26

**Authors:** Tauseef Ahmad, Muhammad Imran, Kabir Ahmad, Muhammad Khan, Mukhtiar Baig, Rami H. Al-Rifai, Basem Al-Omari

**Affiliations:** 1Department of Epidemiology and Health Statistics, School of Public Health, Southeast University, Nanjing 210096, China; tahmad@seu.edu.cn; 2Department of Surgery, Faculty of Medicine in Rabigh, King Abdulaziz University, Jeddah 25289, Saudi Arabia; minmuhammad@kau.edu.sa; 3Liaoning Provincial Key Laboratory of Cerebral Diseases, Department of Physiology, Dalian Medical University, Dalian 116000, China; kbrahmad4@gmail.com; 4Department of Biotechnology and Genetic Engineering, Hazara University Mansehra, Mansehra 21120, Pakistan; muhammadkhan1985@gmail.com; 5Department of Clinical Biochemistry, Faculty of Medicine, Rabigh, King Abdulaziz University, Jeddah 25289, Saudi Arabia; drmukhtiarbaig@yahoo.com; 6Institute of Public Health, College of Medicine and Health Sciences, United Arab Emirate University, Al Ain 15551, United Arab Emirates; rrifai@uaeu.ac.ae; 7Department of Epidemiology and Population Health, College of Medicine and Health Sciences, Khalifa University, Abu Dhabi 127788, United Arab Emirates; 8KU Research and Data Intelligence Support Center (RDISC) AW 8474000331, Khalifa University of Science and Technology, Abu Dhabi 127788, United Arab Emirates

**Keywords:** fascioliasis, bibliometric analysis, research trend, web of science core collection, VOSviewer software

## Abstract

**Simple Summary:**

This study was conducted to further our understanding of the global research outcomes, frontiers, trends, and the most-studied areas in fascioliasis, and it is of specific value for veterinarians, doctors, and researchers. A total of 4165 documents published between 1913 and 2021 from 116 countries were analyzed bibliometrically. The results of the bibliometric analysis indicated that the top 50 institutes publishing research documents regarding fascioliasis are from developed countries. Therefore, it is important that researchers in developing countries seek collaboration with other researchers in developed countries to learn new advancements and effective control strategies for fascioliasis. The researchers in developing countries should seek collaboration mainly with those researchers in the USA, France, England, and Spain who published over 35% of the documents related to fascioliasis.

**Abstract:**

*Background*: Fascioliasis is a zoonotic neglected tropical disease caused by *Fasciola hepatica* and *F. gigantica*. In endemic regions, fascioliasis represents a huge problem in livestock production and significantly threatens public health. The present study was performed to assess the key bibliometric indicators, plot the global research outcome, and strive to find the research frontiers and trends in fascioliasis. *Methods*: A descriptive bibliometric and visualized study was conducted. The data were extracted from the Web of Science Core Collection (WoSCC) database. The WoSCC was searched using key terms covering a wide range of synonyms related to the causative agent (*Fasciola*) and the disease (fascioliasis). The database search was performed for the period from the inception of WoSCC until 3 October 2021. The downloaded data were exported into VOSviewer software version 1.6.17 for Windows to construct co-authorship countries, keywords co-occurrence, bibliographic coupling sources, and citation and documents network visualization. *Results*: A total of 4165 documents were included in this bibliometric analysis. The included documents were published between the years 1913 and 2021 from 116 countries, mainly from the United States of America (USA) (n = 482, 11.6%). The most prolific year was 2018 (n = 108). The journal that attracted the most publications was *Veterinary Parasitology* (n = 324), while the most productive author in this area was Rondelaud D (n = 156). In terms of total link strength (TLS), the most influential country was Spain (TLS = 236), followed by the USA (TLS = 178). *Conclusion*: This study is of value for veterinarians, doctors, and researchers to explore insights into research frontiers and trends in research on fascioliasis. The number of publications on fascioliasis has increased over time. Above 35% of publications have been produced by the USA, France, England, and Spain. “*Fasciola hepatica*” and “cattle” were the most dominant and widely used keywords. Research collaboration should be established among the researchers from developing countries with developed countries to learn new advancements and effective control strategies for fascioliasis.

## 1. Introduction

Fascioliasis is a waterborne and foodborne zoonotic disease caused by the trematode species *Fasciola hepatica* and *F. gigantica* [1]. *F. hepatica* has a worldwide distribution, and it is reported in Asia, Europe, America, Oceania, and Africa, whereas *F. gigantica* is reported in Africa and Asia [1]. In veterinary medicine, the severity of animal fascioliasis affects animals to different extents, depending on the host and parasitic burden. The clinical signs can vary from asymptomatic to a devastating disease (e.g., weight loss, reduced milk production yield, and diminished fertility) including death, which leads to heavy economic losses [2,3,4,5,6].

In the 1980s, human fascioliasis was only considered a secondary disease. However, the increased infection by *F. hepatica* in humans, reported in the 1990s, increased the importance of this disease in public health. [7,8]. To date, human fascioliasis should be considered a significant human parasitic disease [7]. Human infection by infective metacercariae (*trematoda*) can be ingested through freshwater wild plants, freshwater cultivated plants, drinking of contaminated water, and raw liver infected with metacercariae [9]. A recent review [10] has widened the spectrum of human infection sources to include: (1) ingestion of freshwater wild plants, (2) ingestion of freshwater cultivated plants, (3) ingestion of terrestrial wild plants, (4) ingestion of terrestrial cultivated plants, (5) ingestion of traditional local dishes made with contaminated sylvatic plants, (6) ingestion of raw liver, (7) drinking of contaminated water, (8) drinking of beverages and juices made from local plants, (9) ingestion of dishes and soups made with contaminated water, and (10) washing of vegetables, fruits, tubercles, kitchen utensils or other objects with contaminated water.

The long life span of fasciolids in humans is up to 13.5 years [11], which underlies complications and sequelae in long-term chronicity. The most common complications are cholecystitis, biliary obstruction, liver abscesses, subcapsular haemorrhages, and recurrent cholangitis [12]. Other complications include biliary colics, lithiasis, anaemia, and bacteriobilia [13,14,15]. In Turkey, two rare cases of fascioliasis showed tumor-like lesions in the liver [16]. Ectopic fascioliasis can occur in different organs such as the abdominal wall and spleen but not the liver [17]. Others are in the abdominal wall, spleen, pancreas, subcutaneous tissue, blood vessels, heart, lung and pleural cavity, skeletal muscle, epididymis, and appendix [11,18]. The pathogenic and physiological mechanisms underlying neurofascioliasis and ophthalmofascioliasis are restricted to the sporadic cases in which the direct affection of the central nervous system or the eye is caused by a migrant ectopic fasciolid fluke. On the contrary, relatively frequent, or less rare are situations in which fascioliasis infection in the liver gives an indirect rise to neurological affection and similarly are those cases of fascioliasis infection in the liver indirectly causing ocular manifestations [19]. Fascioliasis is easy to diagnose in areas where the infection is endemic; however, the diagnosis is difficult where cases are only detected sporadically [20,21]. In order to understand the global research outcomes, research frontiers, and the most-studied areas in fascioliasis, this bibliometric study was designed. In recent years, bibliometric studies have received growing attention in the medical and health sciences discipline. Bibliometric studies are easy tools to assess and characterize the research output and trends, highly contributing institutions and countries, leading authors and journals, top-cited studies, and other key bibliometric indices [22,23,24,25]. Such information is helpful for researchers, health care professionals and clinicians, veterinarians, and policymakers for better understanding and standard bibliographic information.

## 2. Methods

### 2.1. Study Design

A descriptive bibliometric and visualized study was performed.

### 2.2. Publications’ Duration

The included publications were published from the year 1913 up to 3 October 2021.

### 2.3. Data Source

An online search was conducted in the Science Citation Index-Expanded (SCI-E) Edition of the Web of Science Core Collection (WoSCC) hosted by Clarivate Analytics, Philadelphia, USA. The retrieved database was searched on 3 October 2021. The database was accessed through the electronic library portal of Southeast University, China.

### 2.4. Searching Key Terms and Data Extraction

The Boolean search query method was applied. The searching key terms covered a wide range of synonyms, which includesd “Fascioliasis” OR “*Fasciola hepatica*” OR “*Fasciola-hepatica*” OR “*F. hepatica*” OR “*Fasciola gigantica*” OR “*Fasciola-gigantica*” OR “*F. gigantica*” OR “common liver fluke” OR “sheep liver fluke” in the title field. The retrieved results were further refined by document types. The following document types were excluded: corrections, book reviews and chapters, news items, discussions, and retracted publications. The key data attributes that were extracted included: study title, publication year, journal name, author(s) name(s), keywords, institution, and country. All extracted data including the number of publications and number of citations are based on the WoSCC database.

### 2.5. Network Visualization

The prerequisite network visualization was constructed using VOSviewer software version 1.6.17 for Windows. VOSviewer is a freely available, easy to perform, and widely used visualization landscape tool [24,26,27,28,29]. The exported data were plotted for co-authorship countries, keywords’ co-occurrence, bibliographic coupling and sources, and citation and documents network visualization. For each plotted country, the total link strength (TLS) of the co-authorship countries was calculated. The TLS attribute represents the total strength of the author/researcher links of a given country with other countries. Furthermore, the stronger the link between the two countries, the thicker the line in the network visualization. In addition, each color represents a different cluster. The minimum cluster size was adjusted at ten.

## 3. Results

### 3.1. Publication Growth

After excluding the following documents: corrections (n = 26), book reviews and chapters (n = 14), news items (n = 02), discussions (n = 02), and retracted publications (n = 01), a total of 4,165 documents published between 1913 and 3 October 2021 were included in the final analysis. These documents were cited 66,518 times with an average of 15.97 citations per document and an overall H-index of 86. The most productive year in terms of published documents was 2018 (n = 108, 2.6%), followed by 2015 (n = 107, 2.6%), 2011 (n = 104, 2.5%), then 2020 (n = 102, 2.4%) and, so far, (n = 69, 1.7%) 2021, as shown in Figure 1.

### 3.2. Document Types, Languages, Funding Agencies, and Most-Studied Research Areas

In total, the highest number of published documents were research articles (n = 3378, 81.1%), meeting abstracts (n = 318, 7.6%), and review articles (n = 54, 1.3%), and the remaining documents (n = 415, 10.0%) were either notes, letters, editorial material, or proceeding articles. Most of the documents were published in the English language (n = 3853, 92.5%), and the majority of all included studies were funded by European Commission (n = 162, 3.9%). The most-studied research area was parasitology (n = 1833, 44%), followed by veterinary sciences (n = 1254, 30.1%) and tropical medicine (n = 387, 9.3%). Figure 2 shows the full details.

### 3.3. Top Five Leading Authors, Journals, Institutions, and Countries

The author who published the highest number of documents in fascioliasis research was Rondelaud, D (n = 156, 3.7%), followed by Fairweather, I (n = 129, 3.1%), and Dalton, JP (n = 97, 2.3%). The journals that attracted the highest number of publications were Veterinary Parasitology (n = 324, 7.8%), then Parasitology (n = 237, 5.7%). Approximately 40% of the published documents were from the journals hosted by Elsevier (n = 1222, 29.3%) and Springer Nature (n = 426, 10.2%). Queen’s University Belfast was the most active institution producing published documents related to fascioliasis research (n = 250, 6%), followed by the University of Puerto Rico (n = 112, 2.6%) and the University of Valencia (n = 111, 2.6%). The highly contributing countries were the USA (n = 482, 11.6%), France (n = 333, 8%), and England (n = 330, 7.9%). Figure 3 shows the full details.

### 3.4. Top Ten Most-Cited Papers in Fascioliasis

The most-cited document in fascioliasis research was “Fascioliasis and other plant-borne trematode zoonoses” (Mas-Coma et al., 2005b) with 521 citations. Among the top ten most-cited documents, there were seven articles, two reviews, and one note. Table 1 shows the full details.

### 3.5. Co-Authorship Countries Network Visualization

The minimum number of documents of a country was fixed at five. Of the 116 countries, 65 countries met the threshold and were plotted. Based on publications, the USA was the leading country in terms of published studies, while, based on the TLS, Spain was the most influential country with a TLS of 236, followed by the USA (TLS = 178) then England (176), as shown in Figure 4.

### 3.6. Co-Occurrence All Keywords Network Visualization

The minimum number of occurrences of a keyword was set at 20. Of the total 5560 keywords, 166 keywords met the threshold and were plotted for network visualization. The most widely used keyword was “*F. hepatica*” with a total of 821 occurrences (TLS = 3803), followed by “cattle” (occurrences = 436; TLS = 2,431) and “sheep” (occurrences = 382; TLS = 2160), as shown in Figure 5.

### 3.7. Bibliographic Coupling Sources Network Visualization

The minimum number of documents of a source was ten. Of the total 630 sources, 68 met the threshold and were plotted. The Veterinary Parasitology had the highest TLS 122,834, followed by Parasitology Research (TLS = 66,099) and Parasitology (TLS = 56,295), as shown in Figure 6.

### 3.8. Documents Citation Network Visualization

The minimum number of citations of a document was selected at 50. Of the total documents, only 258 met the threshold and were plotted. Mas-Coma (2005a) was the most cited document, followed by Mas-Coma (1999) and Mas-Coma (2005b), as shown in Figure 7.

## 4. Discussion

Bibliometric analysis plays a significant role by providing the referral point [38]. This study focused on the global research trends and outcomes on a neglected tropical disease (fascioliasis). Our bibliometric analysis documented that there was a continuous growth in fascioliasis-related publications from 1913 to 2021. This growth was irregular, as a very rapid increase in publications was observed from 1965 to 1985. This rapid increase is justified by the deadly outbreak of fascioliasis in Britain and Europe that produced thousands of cases and caused huge economic loss during that period [17,39]. In 2018, the highest number of articles on fascioliasis was published. This could be due to the increased economic reliance on livestock during this period, which subsequently increased the importance of the disease and resulted in an allocation of more funding for research in this area.

The USA has published the most studies followed by France and the UK. The current study has similar trends with many other bibliometric studies in different fields that confirm the USA as a global research leader both quantitative and qualitatively [40,41,42,43,44,45]. On the other hand, Spain had the highest TLS value, followed by the USA and the UK. This unveils the fact that the number of publications does not always lead to receiving more citations, rather, it is the contents, originality, usefulness, and new contributions of the study. One of the top five leading authors in fascioliasis (Santiago Mas-Coma) received the highest citations count for one of their published studies [30]. This is consistent with the literature published in 2016 [46] and suggests that the prominent researchers in the field of *F. hepatica* life cycle are Ben Dawes, Michael VK Sukhdeo, and Santiago Mas-Coma. Queen’s University Belfast was the institution that produced the highest number of published studies on fascioliasis research. In this category, not only in the top ten but the top 50 institutes are from the developed countries. All the leading institutes are well established and renowned with sufficient research funding and well-established research laboratories. Therefore, researchers in developing countries should collaborate with other researchers in developed countries to learn new advancements and effective control strategies for fascioliasis.

Keywords play a vital role to locate any research in locating the required document. In our analysis, the most used keywords were “fascioliasis” and “cattle.” This represents that the disease burden in livestock is high. The first keyword points out the etiology of disease, while the second one shows that cattle are most affected among other animal species. In fascioliasis, most published studies focused on the basic information on prevalence, disease burden, causes, and epidemiology. Few studies focused on the diagnoses and treatment of fascioliasis, which is of key importance to combat the disease. This led to overstudying some aspects of a disease or research problems while neglecting other important aspects like reliable and accurate diagnosis and treatment. This phenomenon may increase the number of articles on a topic while decreasing the number of citations and reach of the articles as similar information is being published in many articles. Contemporarily, the studies addressing some issues of the field are receiving more attention from the researchers and scientific community based on the value of the addressed topic during the time that it is being cited. There should be a balance between both aspects stated above. Sufficient information is necessary to understand the disease for the diagnosis initially and later to the treatment. Over-studying one aspect of a disease loses the interest of the scientific community, which could hinder the control of the intensity and disease burden.

Journals are considered important tools for the dissemination of research; thus, the quality and prestige of a journal plays a major role in transmitting the research to the concerned segment of society [47]. Most of the studies have been published in Veterinary Parasitology (IF: 2.73), Parasitology (IF: 3.23), and Experimental Parasitology (IF: 2.01). Generally, the authors prefer relevancy over the IF of a journal. The specialty of a journal plays an important role in attracting relevant studies from different regions. As fascioliasis is a parasite disease and as it is based on veterinary science, most of the articles were published in journals focused on these aspects, while the journal named “Experimental Parasitology” provided fewer publications amongst the top three journals in fascioliasis, which shows a specific mindset of the authors that chose the journal for publication. However, when Advances in Parasitology celebrated a hundred volumes, they identified a fascioliasis publication (Book chapter) by Mas-Coma and colleagues [1] as one of the ten most-cited papers in Advances in Parasitology [48] with 23.2 average citations per year, compared to the second top article on Anisakiasis with 19.8, which certainly highlights the importance of fascioliasis. Furthermore, the access of the reader to the article also affects the number of citations. This could be due to limited access to the published literature.

Due to limited budgets, the universities usually have a limited subscription. As a result, the scientific community mostly can only reach the open-access article. In the current study, 27.11% of articles all had open access, gold access, and gold-hybrid access. Besides the journals, the publishers also play a vital role. Almost 40% of the retrieved publications on fascioliasis are published in journals hosted by Elsevier and Springer Nature. This might be due to the no-article-processing or publication charges or the publisher’s open- and closed-access policy. Funding agencies and research organizations’ role in promoting science and research is of key importance [49]. The results show that most of the funding agencies were from the USA and other developed countries. The finding of our study is in line with other studies [50].

Our findings highlight the increased research activity on neglected tropical diseases, considered the growing importance in several countries and the need for further studies focused on preventive and therapeutic aspects. The endemic areas should be screened for effective prevention. Moreover, cytology, surgeries, and treatments are the neglected areas that have practical applications and scope of publication and citations as well.

## 5. Conclusions

Our findings are of specific value for veterinarians, doctors, and researchers and explore insights into research frontiers and trends in fascioliasis. This study provided a baseline reference point to devise future policies to effectively manage the disease in endemic areas and future research funding. In fascioliasis, the most widely used keyword was “*F. hepatica*,” while the most-studied research area was parasitology. Above 35% of publications related to fascioliasis have been produced by the USA, France, England, and Spain. Therefore, research collaboration should be established among researchers from developing countries with developed countries to learn new advancements and effective control strategies for fascioliasis.

## 6. Limitations

The current study utilized only one database search (WoSCC), which may have an impact on the citations count and the publication frequency in fascioliasis. However, WoSCC is one of the most common databases used for bibliometric analysis. Our data must be interpreted in line with this limitation.

## Figures and Tables

**Figure 1 animals-11-03385-f001:**
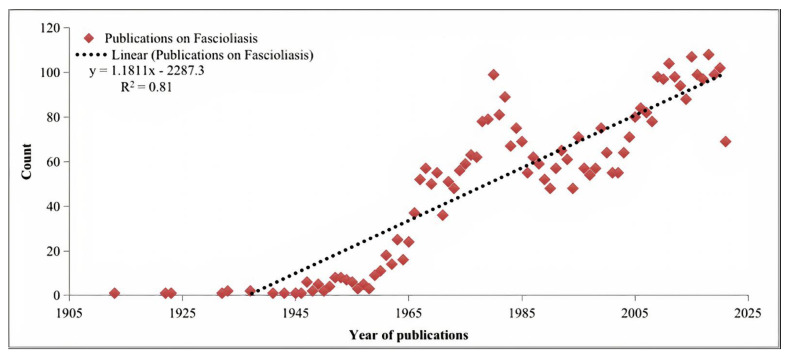
Time trend of publications on fascioliasis from 1903 to 2021.

**Figure 2 animals-11-03385-f002:**
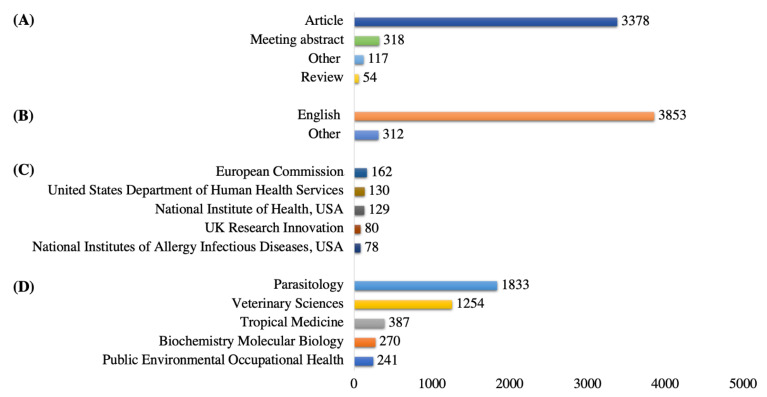
(**A**) Document types, (**B**) publishing languages, (**C**) top five funding agencies, (**D**) top five most-studied research areas in fascioliasis.

**Figure 3 animals-11-03385-f003:**
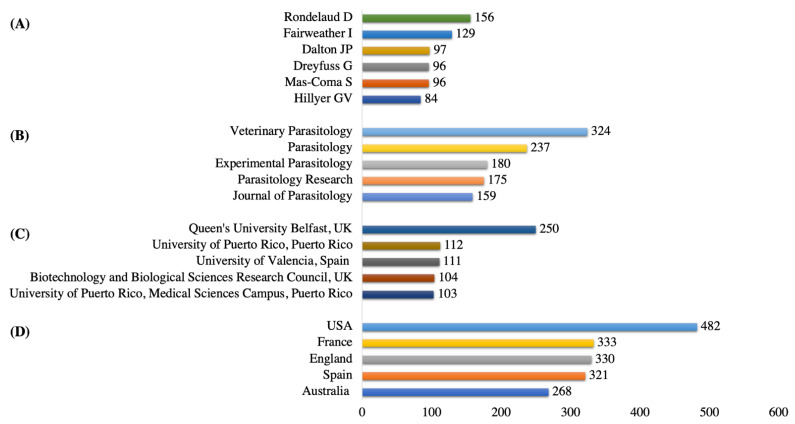
(**A**) Top five authors, (**B**) top five journals, (**C**) top five institutions, (**D**) top five countries in fascioliasis research.

**Figure 4 animals-11-03385-f004:**
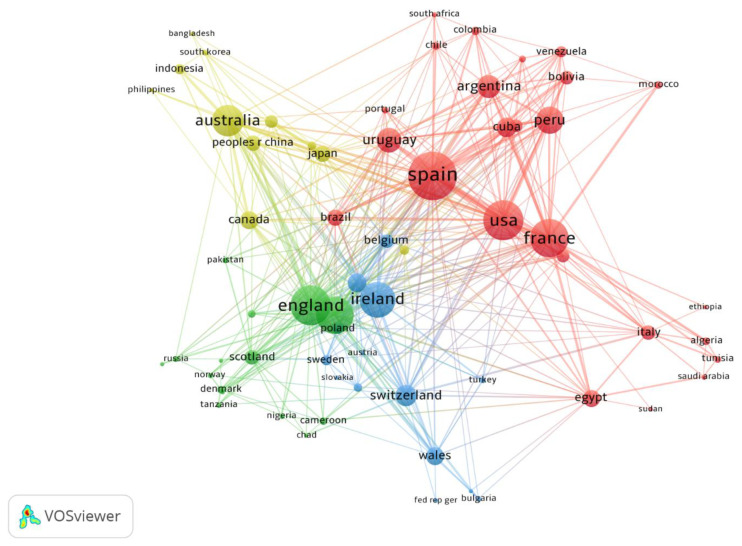
Co-authorship countries network visualization based on TLS.

**Figure 5 animals-11-03385-f005:**
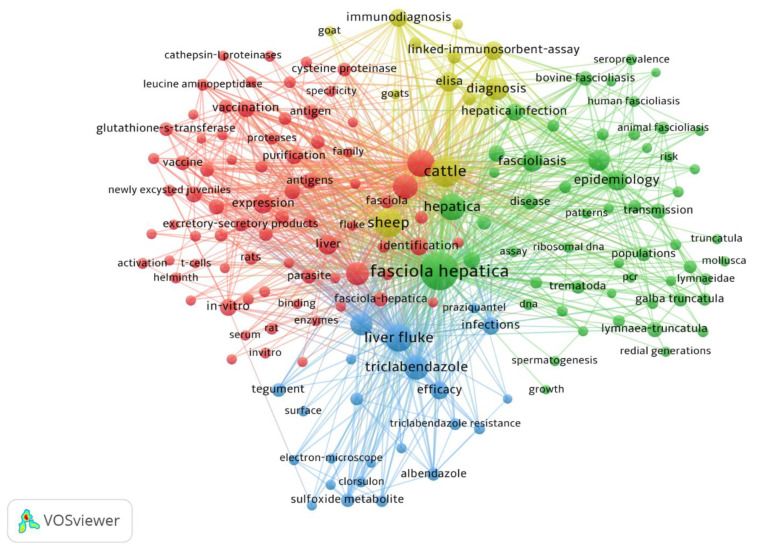
Network visualization of co-occurrences of all keywords based on occurrences.

**Figure 6 animals-11-03385-f006:**
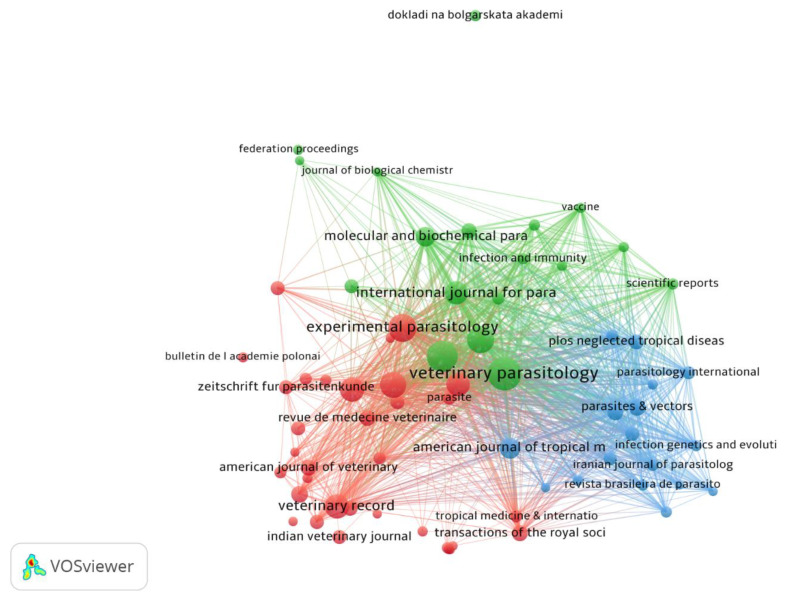
Network visualization of bibliographic coupling sources based on documents.

**Figure 7 animals-11-03385-f007:**
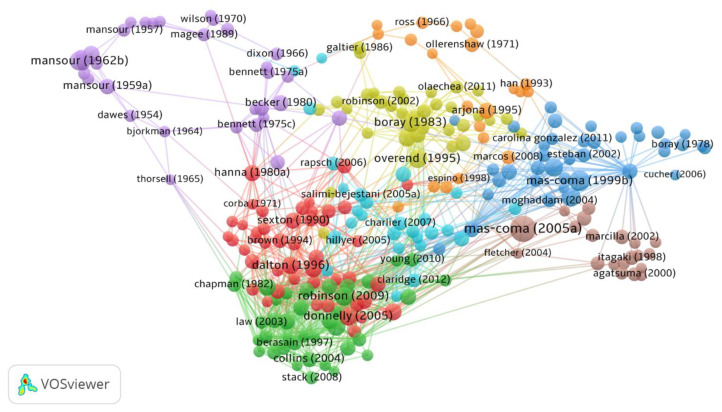
Network visualization of citation and documents based on citations.

**Table 1 animals-11-03385-t001:** Top ten most-cited papers in fascioliasis research.

Raking	Title	Paper Type	Total Citations	Average Per Year	Citations in	
					2020	2021
1st	Fascioliasis and other plant-borne trematode zoonoses [30]	Review	521	30.65	46	21
2nd	Epidemiology of human fascioliasis: a review and proposed new classification [7]	Article	280	12.17	10	5
3rd	Epidemiology of fascioliasis in human endemic areas [8]	Article proceeding paper	224	13.18	22	8
4th	Effects of serotonin (5-hydroxytryptamine) and adenosine 3′,5′-phosphate on phosphofructokinase from liver fluke *Fasciola hepatica* [31]	Article	218	3.63	1	0
5th	Treatment of immature and mature *Fasciola-hepatica* infections in sheep with triclabendazole [32]	Article	210	5.38	8	3
6th	Climate change effects on trematodiases, with emphasis on zoonotic fascioliasis and schistosomiasis [33]	Article proceeding paper	208	16	19	9
7th	Thioredoxin peroxidase secreted by *Fasciola hepatica* induces the alternative activation of macrophages [34]	Article	207	12.18	16	10
8th	An integrated transcriptomics and proteomics analysis of the secretome of the helminth pathogen *Fasciola hepatica* proteins associated with invasion and infection of the mammalian host [35]	Article	198	15.23	19	12
9th	*Fasciola hepatica* cathepsin L-like proteases: biology, function, and potential in the development of first generation liver fluke vaccines [36]	Review	198	10.42	8	8
10th	Resistance of *Fasciola-hepatica* to triclabendazole [37]	Note	197	7.3	8	7

## Data Availability

Not applicable.

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
