# Peer review of "A Bibliometric Analysis and Global Trends in Fascioliasis Research: A Neglected Tropical Disease"

_animals, 2021, doi:10.3390/ani11123385_

Round 1

Reviewer 1 Report

This is a quite original manuscript, written from an unusual point of view. However, it is evident that the medical and veterinary importance of fascioliasis undoubtedly merits such an approach and that readers from fields other than the history of Parasitology, history of diseases, etc., will acknowledge it.

There are however many aspects to be improved before this manuscript can be accepted for publication.

1.- Title: The use of the term "mapping" leads to easy misinterpretations. Change to "Bibliometric approach and global research analysis .....".

2.- Simple Summary: The last sentence lacks a verb.

3.- Abstract: Put species in italics. Delete "mapping in line 44."

4.- Introduction, Line 63: Change <<Fasciola hepatica (F. hepatica) and Fasciola gigantica (F. gigantica)> to <<Fasciola hepatica and F. gigantica>>, with species in italics (apply italics to species throughout).

5.- Lines 75-77:  A recent review has widdened the spectrum of human infection sources. Add the following reference and complete the list of sources according to this article:

Mas-Coma, S.; Bargues, M.D.; Valero, M.A. Human fascioliasis infection sources, their diversity, incidence factors, analytical methods and prevention measures. Parasitology 2018, 145 (13, Special Issue), 1665-1699.

Section 2.5: Again the use of "mapping" should be deleted or substituted by another more appropriate term to avoid confusion with geographical or climatic mapping.

Figure 2: Resolution of the image is not sufficient. Additionally, increase the size by using whole page width.

Section 3.3: Authors should include something about the information about leading authors in the field of fascioliasis highlighted in the following paper:

Moazeni, M.; Ahmadi, A. Controversial aspects of the life cycle of Fasciola hepatica. Experimental Parasitology 2016, 169, 81–89.

Figure 3: Same as for figure 2 above.

Section 3.4 and Table 1: When entering in WoS, it is easy to see that data of the manuscript are pronouncedly outdated, despite authors noting in the manuscript to have made the analyses at the end of October 2021. For instance, the article ranked 1 is noted to have 521 citations, whereas the WoS notes 563 citations for that article. Such a big difference of more than 40 citations in only a few days (between manuscript reception by the journal and reviewer reception) is very difficult to understand. Similarly, the article ranked 2nd by WoS is the one of the Advances in Parasitology which the authors include as No. 1 in their References section but they overlook it in the table. WoS gives 415 citations to this article, which indeed is not a review but an article with plenty of new data, new research contributions, new results and proposals and even new terminology. Moreover, again around a difference of 40 citations appears concerning the article noted as 2nd in the table. Such big differences indicate that this table 1 should be adequately updated.

In this section 3.4, authors should take into account the following article in which it is highlighted that the article published in the famous Advances in Parasitology which had received more citations is one on fascioliasis:

Stothard, J.R.; Rollinson, D. An Important Milestone in Parasitology: Celebrating a Hundred Volumes of Advances in Parasitology. Adv. Parasitol. 2018, 100, 1-27

Figure 4: Same as for figure 2 above.

Section 3.6: Again care with the use of the term "mapping".

Figure 5: Put words inside colour rectangles in black letters to facilitate the reading.

Section 3.7: Again care with the use of the term "mapping".

Figure 6: Put words inside colour rectangles in black letters to facilitate the reading.

Lines 209-210: Correct order of cited articles according to updated data in WoS. See comments on this issue above. I wonder whether the correction of this order may lead to the need of re-assessing the data of figure 7.

Figure 7: Put words inside colour rectangles in black letters to facilitate the reading. Resolution of the image is not sufficient. Additionally, increase the size by using whole page width.

Line 232: The term "nature" is not  appropriate in this sentence because of undefinition. Change to <<contents, originality, usefulness, and new contributions>> or similar words.

Line 250: What do authors understand by the term "burning issues"? Please clarify in the text. Authors should also consider that an issue may be considered burning during a time period but not during other periods.

Line 279: Correct to "considered".

Author Response

The authors would like to thank you for allowing us to revise our manuscript. The authors also would like to thank the reviewer for the comments and recommendations that we believe have improved the quality of the manuscript. The authors have made the required amendments based on your recommendations, please see below point by point responses. All amendments to the manuscript are made in blue font and added references are highlighted in yellow, to facilitate your review.

Reviewer 2 Report

Comments and Suggestions for Authors

The manuscript “Bibliometric analysis and global research mapping of Fascioliasis: a neglected tropical disease” provides useful information on fasciolosis worldwide by focusing on parasitic diseases.

Here are some considerations

INTRODUCTION

Write the names of the parasites in italics (in the introduction as well as in the text) 

Line 63: Parasites should be written in italics. please rewrite. Fasciola hepatica (F. hepatica) and Fasciola gigantica (F. gigantica). Reword in all manuscript.

Line 64: F. hepatica exists… F. hepatica has a wordwide distribution and it is reported in…

Line 64: F. gigantica reword in italics and it is reported in

Line 65: In veterinary medicine,

Line 67: reduced milk production

Line 69-74: I suggest to re-write the sentence. In the 1980s, human fascioliasis was only considered a secondary disease. However, the increased infection by F. hepatica in humans, reported in the 1990s, increased the importance of this disease in public health. To date, human fascioliasis should be considered as a significant human parasitic disease.  

Line 83: I suggest to re-write “Ectopic fascioliasis can occur in different organs such as abdominal wall, spleen…

DISCUSSION

Line 227: I would suggest to delete Result represents that and reword “The USA has published the most of the papers…”

Line 261: As fascioliasis is a parasitic disease

I would probably also improve the discussions by differentiating the pathology in animals and humans

I suggest to improve the quality of the figures as they are not easy to read.

Author Response

(The authors gave the same response as above.)

Round 2

Reviewer 1 Report

Many aspects have already been appropriately corrected but there are still others unavoidably needing to be improved.

1.- Simple Summary, line 34: Change <<documented>> by <<documents>>

2.- Introduction, lines 95-96: Authors confuse the concepts of "neurofascioliasis" and "ophthalmofascioliasis". There are four different situations: The terms of (i) neurofascioliasis and (ii) ophthalmofascioliasis are restricted to the sporadic cases in which the direct affection of the central nervous system or the eye is caused by a migrant ectopic fasciolid fluke. On the contrary, relatively frequent or less rare are situations in which (iii) fascioliasis infection in the liver gives an indirect rise to neurological affection and similarly are those cases of (iv) fascioliasis infection in the liver indirectly causing ocular manifestations.

3.- Section 3.3, line 181: Text and Figure do not fit. I guess text should refer to Figure 3 here.

4.- Discussion, lines 288-289: Authors overlooked tables 1 and 2 in the article of Stothard and Rollinson in Advances in Parasitology (2018). In these two tables detailed data are given on the average citations per year for the best articles appeared in this famous journal, to avoid the impact of time (i.e. older articles have more citations simply because of time). Table 2 clearly notes the article on Fascioliasis as the top best article published by Advances in Parasitology with 23.2 average citations per year, followed at distance by one on Anisakiasis with 19.8 as second. This is crucial because it highlights the importance of Fascioliasis.

5.- Reference No. 9 is not correct. There is only one chapter of this book dealing on fascioliasis. Correct as follows:

Mas-Coma, S. Human fascioliasis. In: World Health Organization (WHO), Waterborne Zoonoses: Identification, Causes and Control; Cotruvo, J.A., Dufour, A., Rees, G., Bartram, J., Carr, R., Cliver, D.O., Craun, G.F., Fayer, R., Gannon, V.P.J., Eds.; IWA Publishing, London, UK, 2004; pp. 305-322.

Author Response

The authors would like to thank you for allowing us to revise our manuscript for the 2nd round. The authors also would like to thank you for the comments and recommendations that we believe are improving the quality of the manuscript. We have made the suggested amendments based on your recommendations, please see the attached file. 
